# The FtsHi Enzymes of *Arabidopsis thaliana*: Pseudo-Proteases with an Important Function

**DOI:** 10.3390/ijms22115917

**Published:** 2021-05-31

**Authors:** Laxmi S. Mishra, Christiane Funk

**Affiliations:** Department of Chemistry, Umeå University, SE-901 87 Umeå, Sweden; laxmi.mishra@umu.se

**Keywords:** AAA-type protease, *Arabidopsis thaliana*, FtsH metalloprotease, chloroplast, embryo lethal, leaf variegation, plastid biogenesis, protein import, oxidative stress

## Abstract

FtsH metalloproteases found in eubacteria, animals, and plants are well-known for their vital role in the maintenance and proteolysis of membrane proteins. Their location is restricted to organelles of endosymbiotic origin, the chloroplasts, and mitochondria. In the model organism *Arabidopsis thaliana*, there are 17 membrane-bound FtsH proteases containing an AAA^+^ (ATPase associated with various cellular activities) and a Zn^2+^ metalloprotease domain. However, in five of those, the zinc-binding motif HEXXH is either mutated (FtsHi1, 2, 4, 5) or completely missing (FtsHi3), rendering these enzymes presumably inactive in proteolysis. Still, homozygous null mutants of the pseudo-proteases FtsHi1, 2, 4, 5 are embryo-lethal. Homozygous *ftshi3* or a weak point mutant in *FTSHi1* are affected in overall plant growth and development. This review will focus on the findings concerning the FtsHi pseudo-proteases and their involvement in protein import, leading to consequences in embryogenesis, seed growth, chloroplast, and leaf development and oxidative stress management.

## 1. Introduction

Cells have evolved an extensive system of molecular chaperones, folding catalysts, and proteases that control protein quality and prevent damage. In addition to the well-studied degradative removal of damaged or superfluous proteins [1], proteolysis is highly important in regulating protein preprocessing, maturation, post-translational protein modifications, and signaling [2,3]. Therefore, it is is no overstatement that proteolysis is directly or indirectly involved in most cellular processes [4].

### 1.1. Proteases in the Plant Chloroplast

Plant proteases associated with a particular proteolytic activity are present in various cellular compartments and organelles constituting up to 3% of the plant proteome [5,6]. Proteases are classified according to their catalytic types. Except for glutamic acid proteases, representatives from all protease classes (threonine, cysteine, serine, aspartic, metalloproteases) have been detected in the plant *Arabidopsis thaliana* [7]. The chloroplast is a unique organelle of the plant cell; absorption and conversion of light energy in the photosynthetic reaction lead to a permanent need for protein turnover (processing and degradation) to adapt to different light conditions. Excess light adsorption further can cause the formation of reactive oxygen species and damage proteins. Therefore, protein quality and quantity controls are essential [8,9]. In addition to photosynthesis, several metabolic reactions happen in the chloroplast, including the biosynthesis of lipids, amino acids, chlorophylls, and carotenoids; therefore, plastidic proteases are vital regulators [9]. More than 20 different families of chloroplast proteases have been detected, with members localized in specific sub-organellar compartments [3,9].

### 1.2. Plant Pseudo-Proteases

In addition to the proteolytically active proteases, members with mutations in their active site attracted the attention of researchers. Despite their putative proteolytically inactivity, many of these pseudo-proteases have essential roles in the cell, ranging from structural proteins via chaperones [10,11] to enzymes with a new function [12,13,14,15,16,17,18,19,20]. In the chloroplast, pseudo-proteases are found in the families of serine Clp-proteases (ClpRs, [21,22,23]) and FtsH metalloproteases (FtsHis [18,24,25,26]). ClpRs of the Clp protease family lack their catalytic triad. These proteolytically inactive subunits are of structural importance to form a tetradecameric proteolytic core together with the catalytically active ClpP subunits [21,22,23]. Most ClpR proteins are essential for the proteolytic function of the Clp core function.

This review will focus on pseudo-proteases belonging to the family of the membrane-bound ATP-dependent FtsH metalloproteases, which are termed FtsHi (*i*-inactive) [23]. These presumably proteolytically inactive FtsHi enzymes are restricted to the plant chloroplast and, similar to the ClpR, affect chloroplast and overall plant development [17,23,24,25].

## 2. Filamentation Temperature-Sensitive Protein H (FtsH) Protease Family

The name FtsH (filamentation temperature-sensitive) erroneously originated from the growth behavior of a Y16 *Escherichia coli* strain deficient in its *ftsh* gene. However, later, a second, independent mutation was found to be responsible for the observed temperature-sensitive phenotype [23].

FtsH proteases, known as zincins, belong to the MEROPS peptidase family M41, which in turn belongs to a larger family of zinc metalloproteases [27]. Within the M41 peptidase domain, the Zn^2+^ ion is ligated by two histidine residues, forming the HEXXH motif (where X is any uncharged residue) as well as a glutamic acid residue [28]. Functional homo- or hetero-hexameric complexes are inserted into the membrane by one or more N-terminal transmembrane domains per subunit [29,30,31,32]. The highly conserved AAA^+^ domain, a cassette of about 200–250 residues that contains the ATP-binding motif (Walker A and Walker B) and the second region of homology (SRH), is situated between the transmembrane region and the active site (Figure 1). Unlike other well-studied ATP-dependent proteases, FtsHs lack robust unfoldase activity [33]. Instead, ATP hydrolysis by FtsH is used to translocate unfolded substrates sequentially into the hexameric pore [33,34]. The AAA^+^ domain is required for nucleotide binding and hydrolysis [29,35] and responsible for alternating between a closed and open state of the FGV pore motif, which is a conserved hydrophobic area at the proteolytic chamber [29]. The substrate is pulled into the degradation chamber via a narrow pore [36,37]. Recent cryo-electron microscopy studies enabled the study of substrate processing of AAA^+^ proteins in detail (reviewed in [38]) and revealed a conserved spiraling organization of ATPase hexamers around the translocating protein substrate.

### 2.1. The FtsH Protease Family of Arabidopsis thaliana

The annual plant *Arabidopsis thaliana* contains 17 different FtsH proteases. Gene comparison studies showed that of the 12 *FTSH* genes potentially coding for fully functional proteases, ten are found in highly homologous pairs. While the pairs AtFtsH1/5, AtFtsH2/8, and AtFtsH7/9 are targeted to the chloroplast, AtFtsH3/10 and AtFtsH4 have been identified in mitochondria. AtFtsH11, the pair partner of AtFtsH4, was initially reported to be dual targeted to mitochondria and the chloroplast [39]. However, Wagner and coworkers confirmed its location to be only in chloroplasts [40]. AtFtsH3 and AtFtsH10 were shown not to be crucial for growth under optimal conditions [41]. Loss of AtFtsH4 leads to oxidative stress and the accumulation of oxidized proteins [42,43]. FtsH10 is involved in the assembly and/or stability of complexes I and V of the mitochondrial oxidative phosphorylation system [43].

#### 2.1.1. FtsH Proteases Located in the Thylakoid Membrane

Of the plastidic FtsHs, FtsH1, 2, 5, and 8 are localized in the thylakoid membrane. These members are the most abundant FtsH proteases and extensively studied [44]; they form hetero-hexameric complexes, in which FtsH1 and FtsH5 (Type A) and FtsH2 and FtsH8 (Type B) can partially substitute for each other [45]. A threshold in the amount of type A and B subunits was postulated to determine the proper function and development of chloroplasts and thylakoid membrane [46,47,48,49]. This thylakoid located protease complex plays a vital role in the degradation and assembly of the Photosystem II reaction center protein D1 and other transmembrane subunits of the photosynthetic machinery. Mutants lacking FtsH2/VAR2 or FtsH5/VAR1 show strongly or slightly variegated leaves, respectively [50,51]. Functional loss of, e.g., FtsH2 results in upregulation of other FtsH proteins in the green leaf sectors to maintain proper function and development of the chloroplasts [47,50,52]. FtsH6 is also localized in the thylakoid membrane. It is essential for thermotolerance and thermomemory in seedlings [53], while no phenotype was observed in adult plants when grown in semi-natural outdoor conditions [54].

#### 2.1.2. FtsH Proteases Located in the Chloroplast Envelope

The other plastidic FtsH enzymes are believed to be localized in the chloroplast envelope [19]. Deleting FtsH7 and 9 does not result in any obvious phenotype [53], and the proteases are not required for PSII repair [55]. FtsH11 is crucial for growth in long photoperiods [40] and thermotolerance [56,57]. FtsH12 was co-immuno-precipitated in a complex with FtsHi1, 2, 4, 5 and NAD-dependent malate dehydrogenase (MDH) and shown to be involved in protein import [58,59]. In addition to FtsHi1, 2, 4, and 5, even FtsHi3 belongs to the five plastidic FtsH homologues, which are incapable of proteolysis in *Arabidopsis*. The FtsHi enzymes either have a mutation in their HEXXH motif (FtsHi1, 2, 4, and 5), or the entire motif is missing (FtsHi3) [18,26]. Compared to AtFtsHi1, 2, 4, and 5, FtsHi3 contains a very short C-terminal domain. Interestingly, FtsHi3 also has undergone a domain swap: the whole M41 domain is located at the N-terminal instead of at the C-terminal to the AAA^+^ domain [26]. Comparing the domain organization of AtFtsHi3 with AtFtsHi1, AtFtsH7, 9, 12 (Figure 1), also AtFtsH7/9 contain this “peptidase M41 FtsH extracellular” domain N-terminal to the AAA^+^ domain, which is additional to their protease domain located in the C-terminal to the AAA^+^ domain. This additional domain is not present in other FtsHs or FtsHis. Whether the N-terminal “peptidase M41 FtsH extracellular” domain of FtsH7, FtsH9, and FtsHi3 enables these enzymes to form a common complex with a specific function remains to be shown. Three independent pre-protein translocating models (pSSU-TEV-protein A, pL11Flag-TEV-Protein A, pLHCP-TEV-protein A) suggested FtsHi3 to form a 1-MD complex separate from the FtsH12/FtsHi1,2,4,5/MDH complex [58] and different to the 1-MD TIC complex [60]. The identity of other components in this complex is unknown. *FTSHi3* is not co-expressed with the tight cluster of *FTSH12/FTSHi1, 2, 4, 5*, but instead with a gene encoding OTP51 [26,58,59]. This pentatricopeptide repeat protein is required for the splicing of group IIa introns and impacts photosystem I and II assembly [61]. The tight co-expression with *FTSHi3* indicates a common function of OTP51 and FtsHi3; therefore, OTP51 is another hypothetical complex partner.

## 3. Pseudo-Proteases with an Important Enzymatic Activity

In addition to being pseudo-proteases, the AAA^+^ domain of FtsHis is intact and—based on the seed lethality of many FtsHi mutants—highly important for their activity, plastid, and overall plant development. Four out of the five AtFtsHi members have been demonstrated to form an inner envelope-bound heteromeric AAA^+^ (ATPase associated with diverse cellular activities) ATPase complex. This complex consisting of FtsH12/FtsHi1,2,4,5/pdNAD-MDH was found to be involved in ATP-driven protein import across the chloroplast envelope [58,59,62,63]. Even Ycf2 was observed being part of this 2 MDa complex using transgenic lines overexpressing *FTSH12* [58]. Still, the protein could not be detected in complexes isolated from wild-type and tic56-3 plastids using a combination of native gel electrophoresis and protein quantification [64] and neither after pull-down of FtsH12 using its native promoter [65]. Kikuchi and coworkers [57] determined the super-complex Ycf2-FtsH12-FtsHi1,2,4,5-pdNAD-MDH physically to interact with TIC components such as Tic214 (Ycf1), Tic100, and Tic56 as well as with the pre-protein translocation components Toc75 and Toc159 [59]. In this complex, neither pdNAD-MDH activity [59] nor FtsH12 proteolytic activity [57] are required; an FtsH12 (H769Y) mutant developed normal chloroplasts with functional pre-protein import abilities.

The finding of ATP-driven protein import across the chloroplast membrane by an FtsH12/FtsHi1,2,4,5/pdNAD-MDH complex has steered an intense debate [66,67] on the importance of the long-accepted chloroplast protein import machinery, i.e., Tic110 and Tic40 forming a common translocon in the inner chloroplast membrane and recruiting the stromal chaperones Hsp93/ClpC1, cpHsp70, and Hsp90C [67]. If indeed the FtsH12/FtsHi complex plays the main role in protein import into the chloroplast [67] remains to be shown. The absence of this complex in most monocots (see Section 4) as well as its lower impact in adult plants (see Section 5) rather point to a specific function during chloroplast development in dicotyledons. We refer the reader to [66,67] and references therein for a detailed description of the subunits involved in protein import.

## 4. Evolution of FtsHi Pseudo-Proteases

Orthologues of the *Escherichia coli* FtsH protease exist in eubacteria and eukaryotic organelles of prokaryotic origin (chloroplasts and mitochondria) but not in Archae. While bacterial genomes contain only one *ftsh* gene, multiple *FTSH* genes are present in photosynthetic organisms, ranging from cyanobacteria to higher plants (reviewed in [18,68]). Even FtsHi enzymes are found already in prokaryotes: the chlorophyll b containing cyanobacterium *Prochlorococcus marinus SS120*, which has a highly reduced genome, contains one gene encoding a presumably inactive version [26]. The evolutionary origin of *ftshi* genes is not resolved yet [15,26,55,58]. Kikuchi and coworkers [58] proposed a progenitor of the chloroplast-encoded Ycf2 to be the predecessor of FtsHi members. Ycf2 has a putative AAA^+^ ATP-binding domain [69] and is essential for cell survival [70]. FtsH1 and FtsH2 were found to associate with Ycf2 in cyanobacteria (*Synechocystis* sp. *PCC 6803*) and Rhodophyta (*Cyanidioschyzon* and *Pyropia*) [58]. Alternatively, bacteria belonging to the Firmicutes were found to have paralogs to the FtsH/FtsHi enzymes acquired via horizontal gene transfer [55].

Based on the finding of an FtsH12/FtsHi1,2,4,5/MDH import complex, we hypothesize a common evolution of multiple FtsHis together with FtsH12 [26]. The protein sequence of AtFtsH12 was blasted against the translated genomes of various species of cyanobacteria, green algae, and higher plants; the BLAST scores, percentage of identity to AtFtsH12, as well as the number of FtsH and FtsHi proteins identified in each species are shown in Figure 2. In *Spirodela* and eudicots, FtsH12 orthologues were identified with an identity of more than 70%. This phylogenetic group also shows the highest copy number of presumably proteolytically inactive FtsH enzymes. The unusually high number of FtsH/FtsHi orthologues found in *Glycine max* is most likely caused by the two duplication events of its genome and several chromosome rearrangements resulting in a palaeopolyploid genome with up to 75% of the genes present in multiple copies [71].

In eudicots, an envelope-located FtsH12–FtsHi complex seems to be necessary for viability. However, two phylogenetic groups, gymnosperms and grasses, have replaced the function of an FtsH12–FtsHi complex. Apart from keeping a high copy number of FtsHs for maintenance of the photosystems and the respiratory chain, these groups contain neither genes with high sequence similarity to FtsH12 nor multiple copies of FtsHi (Figure 2) [62]. In grasses, the FtsH12/FtsHi complex most likely was replaced by an energetically more efficient protein import system that involves Hsp70-type molecular chaperones [66] similar to the mitochondrial protein import system [63].

In early evolutionary plants and cyanobacteria, BLAST scores and percent of amino acid identities to FtsH12 are low (Figure 2). In addition, FtsHi proteins in these organisms are either absent or present in low amounts. Three of the nine FtsH enzymes found in the green alga *Chlamydomonas reinhardtii* are FtsHi pseudo-enzymes [15]. Green algae might have already utilized multiple FtsHs for maintenance and/or import through their inner envelope (Figure 2).

## 5. Phenotypic Consequences

While chloroplast import is a fundamental process throughout the plant´s lifespan, it is interesting to note that reduced levels of FtsHis have a more substantial impact on seedlings than on adult plants. *FTSHi* knock-down mutants display a leaf variegated or pale seedling phenotype, while adult plants look similar to WT [26,72,73,74]. Therefore, import via the FtsH12/FtsHi/MDH complex might be highly important during chloroplast development. These phenotypic consequences further point toward a dose-dependent threshold of single subunits in the FtsH12/FtsHi complex, as observed for FtsHi5 [75]. Studies on the strong and weak *ftshi* mutants may provide insights into the regulatory processes and possible thresholds accountable for a ‘gradient’ of compromised and normal chloroplasts during leaf and chloroplast development.

### 5.1. FtsHi1

*Arabidopsis* ARC1 (accumulation and replication of chloroplasts 1) was isolated by map-based cloning and was found to encode FtsHi1 (At4g23940) [72]. Homozygous *ftshi1-2* knock out mutants are embryo lethal, while the missense mutant *ftshi1-1/arc1* displays a pale-seedling phenotype [72]. Albino seeds in developing siliques of *ftshi1-1/arc1* showed arrest at three different stages of embryo development, namely late globular, early heart, and late heart stage [72]. Seeds harvested from field-grown *ftshi1-1/arc* plants displayed a significant delay in germination compared to their respective WT [26]. The *ftshi1-1/arc1* mutant shows an increased number of chloroplasts, and the plants have smaller rosette sizes throughout their life span [72]. The chloroplast ultrastructure of *ftshi1-1/arc1* showed wavy, swollen, and less organized thylakoids with starch grains accumulating, indicating the chloroplasts still being metabolic active. While chloroplasts of the *ftshi1-1/arc1* mutant contain assembled Ycf2/FtsH12/FtsHi complexes, these plants are impaired in in vitro protein import, which is most likely caused by miss-folding of the AAA-ATPase domain [58]. Mutated FtsHi1 or FtsH12 might partially substitute FtsHi1 in the complex. Gene expression of *FTSH12* and the other *FTSHis* was significantly lower in the *ftshi1-1/arc1* mutant than in WT [26]. Vice versa, the expression of *FTSHi1* was down-regulated in single *FTSHi* mutants [26]. The overexpression of *FTSHi1 (35S: FTSHi1–YFP)* generated few recovered transgenic plants, which were mildly variegated in appearance [72]. These overexpression plants further showed a substantial increase in chloroplast size, but fewer chloroplasts than WT. *FTSHi1* transcript levels were similar in the white and green sectors of the variegated leaves, but in variegated tissue, the level of FtsHi1–YFP was low, while green tissue accumulated FtsHi1–YFP similar to WT [72].

The FtsHi enzymes of the chloroplast envelope have been suggested to respond to photo-oxidative stress [72]. During de-etiolation in white light at intensities of 15, 100, or 300 μmol m^−2^ s^−1^, *ftshi1-1/arc1* mutant seedlings accumulated only half of the chlorophyll amount compared to WT controls. Low light (1 μmol m^−2^ s^−1^) irradiance with 40% blue or 60% red (BR) light led to a significant increase in the mutant’s chlorophyll accumulation. The authors concluded that *ftshi1-1/arc1* at low irradiance still can attenuate chloroplast biogenesis without causing photo-oxidative stress. At normal growth conditions, *ftshi1-1/arc1* displayed lower non-photochemical quenching (NPQ) values than WT. However, during exposure to various stresses (continuous light, long day/high light, short day/4 °C, and long day/30 °C), the mutant sustained similar to WT [26], protection mechanisms might have been triggered [76,77]

### 5.2. FtsHi2

During embryogenesis in plants, a fertilized ovule develops into a plant embryo [78]. Large-scale screens for mutants with altered gametophyte development [75] displayed EMBRYO-DEFECTIVE (EMB) genes that are essential for the growth and overall development of *Arabidopsis* [75,79]. *FTSHi2*, along with *FTSHi4* and *5*, are listed at http://seedgenes.org/index.html (accessed on 16 August 2020) for being EMBRYO-DEFECTIVE (EMB) genes [75,79]. *emb2083-1*, *emb2083-2*, and *emb2083-4* (or *ftshi2-1*, *ftshi2-2*, and *ftshi2-3*, respectively) were investigated by Lu and coworkers [73], and 25% of the ovules in those heterozygous lines were found to be albinos with an embryo-lethal phenotype. Heterozygous *ftshi2* mutants exhibited no evident defects before the globular stage, but then, ≈80% (*n* ≥ 60) were arrested at the globular stage, and the remaining 20% reached the heart-shaped stage with an abnormal division pattern [73]. Consistent with the finding of the FtsH12/FtsHi complex [58,59], FtsHi2 and FtsHi4 were found to interact with each other in vitro [73] and in silico co-expression and qPCR analysis [26]. *FTSH12* and all *FTSHis* co-express with genes encoding enzymes involved in plastid translation, division, and positioning and—with the exception of *FTSHi3*—involved in amino acid metabolism [80].

### 5.3. FtsHi3

FtsHi3 is not associated with the FtsH12/FtsHi1,2,4,5 import complex but instead seems to form a 1-MD complex with unknown partners [58,64]. Further investigations are required to identify its role in the chloroplast envelope. *ftshi3*-KO plants showed residual albino growth in young leaves [58]. The homozygous *ftshi3-2* mutant displays a pale-seedling phenotype when grown for eight days on agar plates, indicating delayed chloroplast and thylakoid membrane development. Six-week-old *ftshi3-2* mutant plants are pale compared to WT when grown in cold stress under short-day conditions [26]. *ftshi3-2* plants displayed reduced Darwinian fitness in comparison to WT [26]. After stress exposure for some days, *ftshi3-2* mutants showed a significant drop in NPQ values than WT but later recovered. After exposure to stress for 6 weeks, the mutant displayed higher NPQ values than the control. Therefore, loss of FtsHi3 might enhance tolerance to photo-oxidative stress and photo-protection [81]. Another homozygous *AtFTSHi3* knock-down mutant *(ftshi3-1(kd))* displayed a significant delay in seed germination compared to WT (Mishra and Funk, unpublished results). This phenotype was attributed to over-accumulation of ABA, while *ftshi3-1 (kd)* seedlings showed partial sensitivity to exogenous ABA. *ftshi3-1 (kd)* plants were drought-tolerant up to 20 days after the irrigation was stopped, whereas wild-type plants wilted after 12 days. Although *ftshi3-1(kd)* displayed a drought-tolerant phenotype in aboveground tissue, its root-associated bacterial community responded to drought (Mishra and Funk, unpublished results).

### 5.4. FtsHi4

In addition to its suborganellar location in the chloroplast envelope [19,58,59], FtsHi4 was also identified as a thylakoid membrane-associated protein [73]. If FtsHi4 indeed is dual-targeted to the envelope and the thylakoid membrane or if this result is due to antibody cross-reaction (as shown for FtsH11 [40]) or impurity of the preparation (in the study, no envelope marker protein was used to examine the purity of the thylakoid fraction [73]) remains to be shown. Ubiquitous transcript levels of *FTSHi4* were detected in all organs of 40-day-old wild-type plants. The lowest *FTSHi4* transcripts were present in roots; transcripts were most abundant in young leaves [73]. *A. thaliana* mutants depleted of FtsHi4 display embryo lethality and disrupted thylakoid formation. Heterozygous *ftshi4/FTSHi4* plants exhibit abnormal division pattern within the same silique, with 80% wild-type embryos reaching maturity and 20% arresting at the heart-shaped stage. Then, albino and green seeds are distributed in developing siliques [73]. These results imply that even FtsHi4 can be substituted in the FtsH12/FtsHi complex. Gene expression of *FTSHi2* and *FTSHi3* was enhanced in homozygous *ftshi4-2* mutants compared to WT [26], while on the protein level, the amount of FtsH12 was slightly diminished in the mutant [65].

Significantly lower numbers of seeds per siliques were observed in *ftshi4/FTSHi4-1* mutants grown under semi-natural conditions. These heterozygous *ftshi4/FTSHi4-1* plants [26] and RNAi-FtsHi4 mutant plants [69] are smaller than WT; their cotyledons have white and yellowish leaves. Six-week-old *ftshi4-2* mutant plants exhibited pale phenotypes compared to WT when exposed to cold stress under short-day conditions [26].

### 5.5. FtsHi5

Similar to the other *FTSHi* mutants, *ftshi5* has a chlorotic seedling phenotype. Under ambient CO_2_ conditions 14-day-old *ftshi5* mutant plants displayed partially impaired thylakoid morphology with reduced density, while the chloroplasts developed normally in high CO_2_ conditions [74]. Using a dexamethasone (DEX)-inducible RNA-interference transgene in *FTSHi5* (DEX: RNAi-*FtsHi5*), Wang and coworkers could induce a dose-dependent albino phenotype in new leaves of *A. thaliana* [74]. Thylakoids in *DEX: RNAi-FtsHi5* plants looked wavy, swollen, and less organized upon DEX induction [74]. RNAi-*FtsHi5* mutant plants also exhibited pale-green leaves upon DEX induction [74].

Transcripts of *FTSHi5* were detected in pre-mature seeds, inflorescences, and young leaves [74]. *FTSHi5pro::GUS* transgenic reporter lines showed the highest *FTSHi5* expression in developing seeds, leaves, and pistils. *FTSHi5* transcription exhibits a circadian rhythm with elevated transcript levels at midday and lower levels at night. Transcription also increased after exposure to high light, while high CO_2_ concentrations had no noticeable effect [74]. Lowered *FTSHi5* expression altered the expression of senescence-related genes and genes encoding enzymes of the oxidation-reduction process. Additionally, *ftshi5-1* plants produced higher levels of H_2_O_2_ and higher amounts of antioxidants to maintain the cellular redox balance [74].

## 6. Conclusions

Early plastid differentiation occurs at the globular-to-heart transition stage during plant embryogenesis [82,83]. The accumulation of chlorophyll in embryos begins at the heart-shaped stage. Therefore, chloroplast biogenesis is associated with embryo development and seedling growth [82,83]. Lack of the plastidic FtsHi proteins affects embryogenesis at the globular–heart transition [72,73,75,79]; therefore, the role of these enzymes is critical during chloroplast biogenesis. While most of the phenotypic characteristics observed in *FTSHi* single mutants can be explained by impaired protein import into the chloroplast, the strong impact during early development is intriguing and should be studied further. Chloroplast development is known to proceed differently in the cotyledons and true leaves in dicotyledonous plants [84,85]. The FtsH12/FtsHi1,2,4,5 complex is absent in grasses, and chloroplast development also proceeds differently in monocotyledonous and dicotyledonous species [84,85]. Whether the role of FtsHi is restricted to import via the FtsH12/Ftshi1,2,4,5 complex or has broader impact remains to be shown. Critical evaluation of all available data is necessary to review or extend our current models. Modern techniques, e.g., cryo-EM, should elucidate the comprehensive molecular structures and underlying mechanisms of the TOC-TIC-Ycf2/FtsHi motor complexes. The various weak and strong FtsHi protease mutants might be perfect tools to answer open questions.

## Figures and Tables

**Figure 1 ijms-22-05917-f001:**
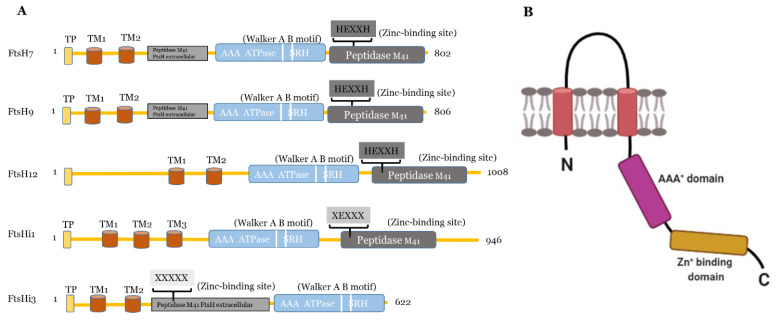
(**A**) Predicted domains and motifs of AtFtsHis in comparison to the presumably active AtFtsH7, 9, and 12. TP, transit peptide; transmembrane domains, TM1-3; Walker A B motifs are indicated as two white lines between the AAA^+^ ATPase and SRH; SRH, second region of homology. Active FtsH proteases contain the Zn^2+^-binding motif (HEXXH) in the their peptidase M41 domain, which is substituted or absent in presumably inactive FtsHis (substitution of both histidines indicated as XEXXX). In FtsHi3, the peptidase M41 domain is annotated as “FtsH extracellular” and is located at the N-terminal to the ATPase; its HEXXH motif is completely missing (XXXXX). FtsH7 and FtsH9 contain an FtsH extracellular” peptidase domain additionally to their protease domain. AtFtsHis are predicted to contain three (FtsHi1, 5), two (FtsHi2, 3), or one (FtsHi4) transmembrane domains (http://www.cbs.dtu.dk/services/TMHMM/, accessed on 3 April 2021) (Supplementary Figure S1 [26]). (**B**) Schematic drawing of the structure of a monomeric FtsH protease with two membrane-spanning regions (shown in red), the proteolytic domain (in range) and the AAA^+^-domain (in pink). Created with BioRender.com (accessed on 27 May 2021).

**Figure 2 ijms-22-05917-f002:**
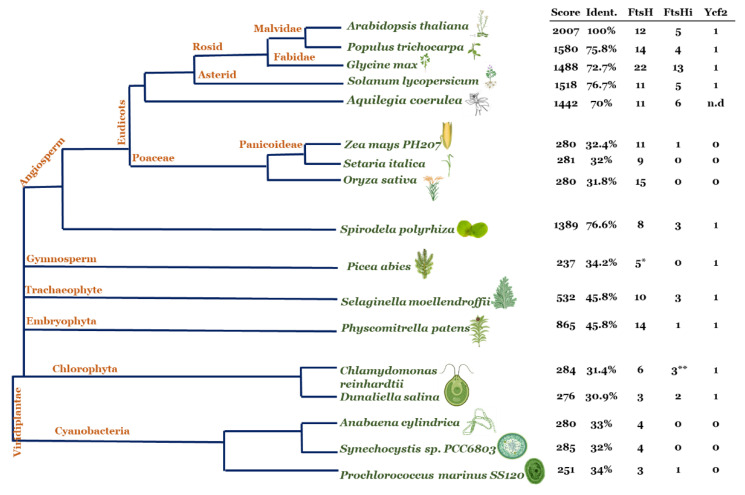
Co-evolution of FtsH12 and FtsHi. The AtFtsH12 amino acid sequence was blasted via phytozome (https://phytozome.jgi.doe.gov/, accessed on 16 August 2018). Hits with the highest score and identity to Arabidopsis FtsH12 are shown at the right hand of the species tree. Furthermore, the number of proteolytic active FtsH and inactive FtsHi in these species were determined by manually investigating the presence of an AAA-like domain and peptidase M41 domain. Sequences containing a zinc-binding motif (HEXXH) within M41 were assumed to correspond to active proteolytic proteases. Data for *Picea abies* were additionally retrieved from congenie.org (marked by *). Data for *Prochlorococcus*, *Synechocystis*, and *Anabaena* were retrieved from https://www.ncbi.nlm.nih.gov/ (accessed on 16 August 2018). Data for *Chlamydomonas reinhardtii* and *Oryza sativa* were supplemented with literature data ([15,68] marked by **). The search for the plastid-encoded Ycf2 was performed in the NCBI organelle genome resources, the chloroplast genome of *Aquilegia coerulea* is not available yet (marked as n.d.). Adapted from [26], created with BioRender.com.

## Data Availability

Not applicable.

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
