# Peer review of "The FtsHi Enzymes of Arabidopsis thaliana: Pseudo-Proteases with an Important Function"

_ijms, 2021, doi:10.3390/ijms22115917_

Round 1
Reviewer 1 Report
The review paper is focusing on FtsHi, pseudo-proteases in Arabidopsis thaliana. The reviewer felt that the paper contains interesting contents. I would like to suggest some points to be improved.
1) Can you add information of FtsHi2, 4, and 5 in Figure 1? The authors describes details in section 5. Or are they the same structure as FtsHi1? If so, please write it in the figure legend.
2) Section 4, Evolution. Can the authors discuss deeply how FtsHi is generated from FtsH? Can the authors explain why Glycine max has doubled number of FtsH and FtsHi?
3) Figure 2. Please explain about asterisks and n.d. (not zero?) in the figure legend.
4) Figures should be more larger because the letters are small.
Author Response
The review paper is focusing on FtsHi, pseudo-proteases in Arabidopsis thaliana. The reviewer felt that the paper contains interesting contents. I would like to suggest some points to be improved.
1) Can you add information of FtsHi2, 4, and 5 in Figure 1? The authors describes details in section 5. Or are they the same structure as FtsHi1? If so, please write it in the figure legend.
We thank the reviewer for this comment. We now have changed the figure legend of Figure 1 and gave a reference to a figure showing the structures of FtsHi2, 4 and 5.
2) Section 4, Evolution. Can the authors discuss deeply how FtsHi is generated from FtsH?
Unfortunately the evolution of FtsHi is not known. As described in lines 185-189 of the manuscript they are hypothesized to have evolved from Ycf2 or alternatively via horizontal gene transfer. Both hypotheses are highly speculative.
3) Can the authors explain why Glycine max has doubled number of FtsH and FtsHi?
We appreciate this comment. An explanation was added to the section.
4) Figure 2. Please explain about asterisks and n.d. (not zero?) in the figure legend.
We thank the reviewer for this improving comment. The figure legend has been changed accordingly.
5) Figures should be more larger because the letters are small.
Figure 2 now has been changed in the hope to improve it.
Reviewer 2 Report
The manuscript reviews and discusses recent literature on plant FtsHi enzymes with the special focus on chloroplasts. First, the FtsH protease family is introduced, and then their properties and evolution discussed. Finally, the phenotypic consequences of (knock-down) mutations are presented. The manuscript is clear and well written, and it gives a comprehensive view to the FtsHi proteases in Arabidopsis thaliana. The manuscript will be of importance for the researchers interested in proteolysis, chloroplast and plant development and chloroplast protein import as well as on chloroplast metabolism.
I have few suggestions that might improve the manuscript:
-lines 34-37: “…absorption and conversion of light energy in the photosynthetic reaction lead to a permanent need for protein turnover (processing and degradation).” Please explain how and why. More detailed explanation is also important to understand section 2.1.1, in which you discuss PSII degradation and assembly.
-lines 39-40. Please also show the protease families in a table.
-extensive use of semicolons in chapter 1.1, please modify the sentences
-lines 61-77: the structure of the protein is difficult to understand just by reading the text. Refer to Figure 1. Moreover, Figure 1 could be extended by presenting a structural model .
-lines 80-90. Phylogenetic tree (showing also the protein locations) would be useful for the reader
-Sentence (lines 259-263) starting with “ftshi1-1/arc1 displayed…” is very complicated, please rephrase.
-lines 318-321. Clarify DEX/DEX inducible RNAi with 1-2 sentences.
-Figures: Resolution was not very good and the texts were not easily seen.
-Typos: line 123, "enableS"; sometimes no spacing between the reference and text (e.g. lines 157, 168), missing italics in several places (see e.g. chapter 4 for for the names of species, “in vitro” in line 243 etc), use of superscripts (lines 255-256). Please check throughout the manuscript.
Author Response
We thank the reviewer for his positive evaluation and answer all comments point by point.
-lines 34-37: “…absorption and conversion of light energy in the photosynthetic reaction lead to a permanent need for protein turnover (processing and degradation).” Please explain how and why. More detailed explanation is also important to understand section 2.1.1, in which you discuss PSII degradation and assembly.
More information was added to the text. However, as PSII degradation and assembly as well as the D1 turnover are regulated by “active”, thylakoid-localized FtsH proteases and not by the envelope-located pseudo-proteases FtsHi, we chose to keep this information short.
-lines 39-40. Please also show the protease families in a table.
While this information is highly interesting, it is not the subject of our review. We therefore chose to cite other reviews dealing with chloroplast-located proteases.
-extensive use of semicolons in chapter 1.1, please modify the sentences
We thank the reviewer for this improving comment and modified the sentences.
-lines 61-77: the structure of the protein is difficult to understand just by reading the text. Refer to Figure 1. Moreover, Figure 1 could be extended by presenting a structural model .
We thank the reviewer for this highly valuable comment. We now refer to Figure 1 in this paragraph and have added a Figure 1b presenting a structural model.
-lines 80-90. Phylogenetic tree (showing also the protein locations) would be useful for the reader
We agree with the reader that such a tree is very useful. However, it has been published several times before by us and others and therefore will not add to this review. It can be found in e.g.:
Shao, S.; Cardona, T.; Nixon, P., Early emergence of the FtsH proteases involved in photosystem II repair. Photosynthetica 2018, 56, (1), 163-177.
Kikuchi, S.; Asakura, Y.; Imai, M.; Nakahira, Y.; Kotani, Y.; Hashiguchi, Y.; Nakai, Y.; Takafuji, K.; Bédard, J.; Hirabayashi-Ishioka, Y., A Ycf2-FtsHi heteromeric AAA-ATPase complex is required for chloroplast protein import. The Plant Cell 2018, 30, (11), 2677-2703.
García-Lorenzo, M.; Pružinská, A.; Funk, C., ATP-dependent proteases in the chloroplast. In: Kutejova E, ed. ATP-dependent proteases. Kerala, India: Research Signpost 2008, 145–176.
García-Lorenzo, M.; Sjödin, A.; Jansson, S.; Funk, C., Protease gene families in Populus and Arabidopsis. BMC Plant Biology 2006, 6, (1), 30.
-Sentence (lines 259-263) starting with “ftshi1-1/arc1 displayed…” is very complicated, please rephrase.
We thank the reviewer for this comment and changed that sentence.
-lines 318-321. Clarify DEX/DEX inducible RNAi with 1-2 sentences.
We acknowledge this comment and changed the sentence accordingly.
-Figures: Resolution was not very good and the texts were not easily seen.
Figure 2 now has been changed and we hope it is easier to read. For submission we also will upload Figure 1 and 2 separately for better resolution.
-Typos: line 123, "enableS"; sometimes no spacing between the reference and text (e.g. lines 157, 168), missing italics in several places (see e.g. chapter 4 for for the names of species, “in vitro” in line 243 etc), use of superscripts (lines 255-256). Please check throughout the manuscript. Read through.
We thank the reviewer for this comment and carefully checked the whole manuscript again.